# Sarcoidosis and COVID-19: At the Cross-Road between Immunopathology and Clinical Manifestation

**DOI:** 10.3390/biomedicines10102525

**Published:** 2022-10-09

**Authors:** Claudio Tana, Francesco Cinetto, Cesare Mantini, Nicol Bernardinello, Marco Tana, Fabrizio Ricci, Andrea Ticinesi, Tiziana Meschi, Riccardo Scarpa, Francesco Cipollone, Maria Adele Giamberardino, Paolo Spagnolo

**Affiliations:** 1COVID-19 and Geriatrics Clinic, SS. Annunziata Hospital of Chieti, 66100 Chieti, Italy; 2Rare Diseases Referral Center, Internal Medicine 1, Ca’ Foncello Hospital-AULSS2 Marca Trevigiana and Department of Medicine-DIMED, University of Padova, 35128 Padova, Italy; 3Department of Neuroscience, Imaging and Clinical Sciences, “G. D’Annunzio” University of Chieti-Pescara, 66100 Chieti, Italy; 4Respiratory Disease Unit, Department of Cardiac, Thoracic, Vascular Sciences and Public Health, University of Padova, 35128 Padova, Italy; 5COVID-19 and Internal Medicine Unit, SS. Annunziata Hospital of Chieti, 66100 Chieti, Italy; 6Internal Medicine Unit, Geriatric-Rehabilitation Department, Azienda Ospedaliero-Universitaria Di Parma, Via Antonio Gramsci 14, 43126 Parma, Italy; 7Medical Clinic, SS. Annunziata Hospital of Chieti, Department of Medicine and Science of Aging, “G. D’Annunzio” University of Chieti-Pescara, 66100 Chieti, Italy; 8COVID-19 and Geriatrics Clinic, SS. Annunziata Hospital of Chieti, Department of Medicine and Science of Aging and CAST, “G. D’Annunzio” University of Chieti-Pescara, 66100 Chieti, Italy

**Keywords:** COVID-19, SARS-CoV-2, sarcoidosis, granuloma, immunopathology, cell, translational medicine, imaging

## Abstract

Coronavirus disease 2019 (COVID-19) has been associated with dysregulation of the immune system featuring inappropriate immune responses, exacerbation of inflammatory responses, and multiple organ dysfunction syndrome in patients with severe disease. Sarcoidosis, also known as Besnier–Boeck–Schaumann disease, is an idiopathic granulomatous multisystem disease characterized by dense epithelioid non-necrotizing lesions with varying degrees of lymphocytic inflammation. These two diseases have similar clinical manifestations and may influence each other at multiple levels, eventually affecting their clinical courses and prognosis. Notably, sarcoidosis patients are at high risk of severe COVID-19 pneumonia because of the underlying lung disease and chronic immunosuppressive treatment. In this narrative review, we will discuss interactions between sarcoidosis and COVID-19 in terms of clinical manifestations, treatment, and pathogenesis, including the role of the dysregulated renin–angiotensin system, altered immune responses involving increased cytokine levels and immune system hyperactivation, and cellular death pathways.

## 1. Introduction

In the last years, the diffusion of a novel coronavirus disease (COVID-19) and its harmful effects on organs, such as the heart and lungs, has been associated with excess mortality and risk of hospitalization [1,2]. The introduction of an effective vaccination campaign has resulted in significant protection against the hyperinflammatory immune response and organ damage from COVID-19, albeit at the moment, there is no information about the long-term efficacy of the vaccine in the global population. The use of booster doses has been introduced in order to reduce waning immunity and immune evasion from SARS-CoV-2 variants; however, there is evidence that there may be a relative reduction of protection over time, in particular in multimorbidity patients and those aged > 65 years [3].

Furthermore, it is even less is known about the clinical and immune interaction between COVID-19 and other disorders [4], in particular those with lung predilection, such as sarcoidosis. Shared immunity, clinical phenotypes, and risk of clinical deterioration in patients with both conditions are still fields of uncertainty [5,6]. This review summarizes the most recent evidence on the relationship between sarcoidosis and COVID-19; in particular aspects of immunopathology, clinical, and diagnostic features are described. Ultimately, clinical algorithms and therapeutic issues at the cross-road between sarcoidosis and COVID-19 will be discussed.

## 2. Immunopathology of Sarcoidosis and COVID-19, Shared Mechanisms

As with other immune-mediated diseases, during the COVID-19 pandemic, sarcoidosis patients have been considered an at-risk population due to their underlying genetic and immunological background [7], possibly impaired lung function, and the immune-suppressive treatment often ongoing in those patients with severe disease. Even before the COVID-19 pandemic, the possible etiologic relationship between sarcoidosis and exogenous/infectious triggers had been extensively discussed [8,9]. Not surprisingly, possibly SARS-CoV-2-related sarcoidosis onset or re-exacerbation has been reported [10,11,12]. The current knowledge about the SARS-CoV-2 infection model may help to find a possible explanation for the above-mentioned clinical observations. A recent analysis of the SARS-CoV-2–host interaction network by time-course multi-omics profiling identified virus-driven profound changes in mRNA expression, protein abundance, ubiquitination, and phosphorylation activities of host cells, providing data about potential common pathways between sarcoidosis and COVID-19 [13]. Moreover, despite T lymphocytes being naturally considered the main players of a viral-driven inflammatory response, the role of humoral and cellular innate immunity has been recently disclosed in anti-viral and pro-inflammatory responses to SARS-CoV-2 [14]. Innate and adaptive immunity are both implicated also in the pathogenesis of sarcoidosis [15].

### 2.1. Adaptive and Innate Immunity in COVID-19 and Sarcoidosis

Parallel to sarcoidosis, lymphopenia has been frequently reported in COVID-19 patients. Although in sarcoidosis, this is considered a general feature of active disease [16], in COVID-19, a low lymphocyte count has been considered an indicator of severe disease and poor prognosis [17,18,19].

The precise mechanisms underlying this observation in COVID-19 patients have not been clarified, but they may include direct lymphocyte killing upon SARS-CoV-2 cellular infection and lymphocyte apoptosis during cytokine storm [20]. Moreover, as for sarcoidosis, lymphopenia could also reflect differences in lymphocyte homing from the peripheral blood to target tissues.

Interestingly, lymphocytes were found to express Angiotensin Converting Enzyme (ACE) 2, and this membrane-bound enzyme represents an essential and major receptor for SARS-CoV-2 to enter the human body host cells, similar to what has been reported for SARS [21]. Dysregulation of the renin–angiotensin system might also be considered a shared point between the two diseases. An increase in ACE has been historically related to granuloma activity [22,23] and is still mentioned among the biochemical features supporting the diagnosis of sarcoidosis [24]. SARS-CoV-2 binding to ACE2 might then reduce Angiotensin II (AngII) inactivation, and the excess of AngII may lead to increased pulmonary vascular permeability and further accumulation of extra-alveolar fluid [25,26]. Of note, Ang II acts as a strong stimulator of apoptosis and as a negative regulator of autophagy, a cell death process that has been involved in the pathogenesis of different neoplastic and immune-mediated conditions [27] SARS-CoV-2 has been shown to specifically increase ubiquitination on different autophagy-related factors (such as MAP1LC3A, GABARAP, VPS33A, and VAMP8) [13].

Moreover, it has been reported that the SARS-CoV-2 spike protein can indirectly activate the autophagy process through the AMP kinase-mTOR pathway [28]. As occurs during SARS-CoV-2 infection, the perturbation of autophagy via the mTOR and Janus kinase (JAK)-1 pathway has also been reported in the pathogenesis of non-Lofgren sarcoidosis [29,30,31].

The role of humoral innate immunity in SARS-CoV-2 response has been deeply investigated, showing that specific humoral fluid-phase pattern-recognition molecules (PRMs) with ancestral antibody-like properties, including collectins, ficolins, pentraxins, and C1q, may play a role both in resistance to and in the pathogenesis of COVID-19 [14]. Circulating and lung myelomonocytic cells and endothelial cells have been reported as major sources of the pentraxin long pentraxin 3 (PTX3) in COVID-19 [32]. PTX3 is a PRM playing essential functions in resistance to pathogens, tissue remodeling, and resolution of inflammation. Notably, in COVID-19, its plasma concentration has been suggested as an independent, strong prognostic indicator of short-term mortality. Interestingly, different reports have hypothesized mechanistic links between the functional activity of humoral innate immunity and sarcoidosis pathogenesis. Significant enrichment in complement-activating factors in bronchoalveolar lavage (BAL) of sarcoidosis patients and increased alveolar concentrations of C5a were detected compared to other interstitial lung diseases. An increased expression of complement receptors has also been reported in monocytes from sarcoidosis patients [33]. Recently, it has been demonstrated that PTX3, in particular, may restrain the pathogenic activation of complement and its downstream impact on metabolic reprogramming of macrophages in sarcoidosis, thus acting as a physiological brake in granuloma formation [34].

Furthermore, acute COVID-19 pneumonia is characterized by an extreme and uncontrolled release of pro-inflammatory cytokines, which implies hyperactivation and dysregulation of both the innate and adaptive immunity, potentially leading to ARDS and to a life-threatening systemic inflammatory response syndrome, known as SIRS [35]. The above-mentioned cytokine storm involves different mediators, including mainly IFN-g, TNF-a, and IL-6, together with G-CSF, IL-1b, IL-1RA, IL-2, IL-4, IL-7, IL-10, and IL-19 [36,37]. IFNg has been found to be a cornerstone/key mediator in active sarcoid disease, and the main cellular source has been identified in tissue-resident macrophages. Additional type 1 cytokines produced by distinct cell types, including IL-6, IL-12, IL-15, and GM-CSF, as well as Type 2 cytokines such as IL-4 and IL-13, have also been associated with the granulomatous microenvironment [38]. Interestingly, many of the listed cytokines implicated in sarcoidosis pathogenesis, including IFN-g and IL-2, IL-23, IL-4, and IL-13, act via the JAK-signal transducer and activator of transcription (STAT) pathway. As a confirmation, JAK-STAT pathway activation has been reported in patients with sarcoidosis [39], and JAK inhibitors represent one of the new therapeutic options for long-standing sarcoidosis [40].

Of note, anti-IL-1b and anti-IL-6 monoclonal antibodies, as well as JAK inhibitors, have been approved for COVID-19 treatment in hospitalized patients [41,42,43,44,45].

### 2.2. Fibrosis

Significant collagen deposition and fibrosis have been observed in 10–20% of chronic sarcoid interstitial lung disease [46]. Mechanisms underlying the transition from inflammatory to fibrotic disease in the lung of sarcoidosis patients are not entirely elucidated, but this complex process is at least in part sustained by increased production of fibrosis-stimulating cytokines (such as TNF-a, IFN-g, TGF-b, IL-10, IL-13) in the granuloma microenvironment [47]. Fibrotic changes have also been reported in COVID-19 pneumonia, even in the early phase of the disease [48,49].

Experimental data from a SARS-CoV-2 in vitro infection model confirmed that SARS-CoV-2 might regulate TGF-b signaling, a multifunctional cytokine playing a key role in the process of tissue repair following injury, by inducing higher expression of FN1 and SERPINE1 [13]. SERPINE1, a member of the serine proteinase inhibitor superfamily, has been shown to promote fibrosis in multiple organ systems and functions as a component of innate anti-viral immunity. Interestingly, among highly active sarcoidosis patients, hypoxia may promote a pro-inflammatory response and a profibrotic response (TGFβ1, PDGF-BB) with SERPINE1 secretion associated with human lung fibroblast migration inhibition [50].

Interestingly, an ongoing trial (NCT04948203) has been designed to evaluate the efficacy of the mTOR inhibitor sirolimus in preventing post-COVID-19 pulmonary fibrosis.

## 3. Clinical Features of Sarcoidosis and COVID-19

### 3.1. Clinical Manifestation of Sarcoidosis

In sarcoidosis, the lung and mediastinal lymph nodes are involved in as many as 90% of patients. Consequently, respiratory symptoms prevail, with dry cough, dyspnea, reduced exercise tolerance, and chest discomfort present in 30–50% of cases [51,52]. Dyspnea is more common in patients with fibrotic lung disease or respiratory tract involvement; however, unlike other interstitial lung diseases, respiratory symptoms may not reflect the severity of the disease extent on chest imaging or functional impairment [53]. Chronic cough and chest pain are two frequent and significant additional symptoms that can contribute to breathlessness, thus limiting exercise activity.

Sarcoidosis may affect virtually any organ, and this is reflected by the wide range of potential organ-specific manifestations [54]. However, non-organ-related symptoms, such as fatigue, fever, memory loss, sleepiness, and dizziness, are also frequently reported. They may be disabling and mimic SARS-CoV2 infection. However, sarcoidosis is asymptomatic in many cases and is often detected incidentally.

Fatigue is one of the most common complaints in both sarcoidosis and COVID-19. In chronic sarcoidosis, the prevalence of chronic fatigue can be as high as 60–70%, and about 25% of patients may manifest severe fatigue, which significantly reduces their quality of life and work productivity [55]. The mechanisms involved in the development of fatigue are largely unknown, but diffuse granulomatous inflammation and systemic cytokine release appear to contribute only marginally to it. Several comorbidities associated with sarcoidosis, including hypothyroidism, sleep apnea, and depression, can either contribute to or be the main driver of fatigue and should be carefully excluded or appropriately managed. Fever is another nonspecific symptom and is common in the acute phase of both sarcoidosis, particularly in patients presenting with Löfgren syndrome, and COVID-19 disease; however, in most sarcoidosis cases, fever is mild and tends to respond to steroid treatment. Small fiber neuropathy (SFN) is a neurological disorder that affects small fibers of the peripheral nervous system. SFN is often associated with sarcoidosis and may represent a major source of frustration for patients. As many as 40–50% of sarcoidosis patients may manifest SFN-related symptoms, including cutaneous hyperalgesia, facial flushing, urinary difficulties, bowel constipation, chronic pain, and palpitations [56,57].

### 3.2. Clinical Manifestations of COVID-19

Among SARS-CoV2 patients, approximately one-half remain asymptomatic and with a self-limited disease course [58]. Conversely, the spectrum of clinical manifestations in symptomatic patients varies widely accordingly to disease severity and progression. In almost all COVID-19 studies, fever is the most frequent manifestation during hospital stay (83–98%) [59,60]. Indeed, since the beginning of the pandemic, fever, which is typically intermitting and can last up to ~14 days, has been considered an important criterion in the definition of suspected cases. Additional common symptoms include fatigue (70%), dry cough (59%), and dyspnea (31%). The range of less common symptoms associated with COVID-19 is much wider and includes headache, myalgia, anorexia, sore throat, chest pain, nausea, diarrhea, vomiting, ageusia, and anosmia [61,62]. The disease generally manifests following an incubation period of 1–14 days (with a median of 5 days), while pneumonia and respiratory failure tend to occur within an average time of 10 days from disease onset [63]. Dyspnea and fever are associated with a more severe disease course; conversely, the prevalence of other symptoms, such as cough, nausea, headache, sore throat, and expectoration, is similar in patients with severe and non-severe disease [64].

Following discharge, up to 30% of COVID-19 survivors may experience persistent symptoms (post-acute COVID-19 syndrome) such as fatigue, joint pain, cognitive disorders, and weight loss [65], which are common symptoms also in sarcoidosis. Additional overlapping manifestations observed in patients recovering from COVID-19 infection include, among others, tachycardia, flushing, and urinary urgency. Interestingly, in a recent case series, signs of Small Fiber Neuropathy (SFN) were observed in the skin biopsies of six subjects after COVID-19 infection [66]; however, following COVID-19 infection, sarcoid-like subcutaneous nodules have also been reported [12]. These findings might represent another potential link between sarcoidosis and COVID-19.

### 3.3. Clinical Features of Sarcoid Patients Infected by SARS-CoV2

Little is known about the long-term consequences of SARS-CoV2 infection in patients with pre-existing chronic lung diseases. Only a few studies have analyzed the impact of SARS-CoV2 infection in patients with sarcoidosis, and data are mainly limited to small case series or isolated case reports, including patients infected in the first year of the pandemic. With these limitations, sarcoidosis patients do not seem to have a higher risk of infection compared to the general population, but a higher risk of severe disease has been reported in a few cohorts. In a study from Spain, the prevalence of SARS-CoV2 infection in patients with sarcoidosis was 5.1%, with 80% of cases being symptomatic. The most frequently reported symptoms are cough (67%), fever (61%), fatigue (42%), and myalgias (28%); conversely, only a minority of patients had dyspnea (22%). Moreover, 31% of patients were hospitalized, and 9% died following COVID-19 infection [67]. During the first pandemic wave, a similar prevalence was reported in a small case series of African American patients in the US (2.1%), with a mortality of 20% in 5 infected patients [68].

In a larger cohort of sarcoidosis patients (*n* = 886), Baughman and colleagues reported a prevalence of infection of 8.9% and a hospitalization rate of 25%, with only one death from COVID-19 [69]. Furthermore, no individual immunosuppressive therapy was associated with an increased risk of SARS-CoV2 infection. Similarly, in a French cohort of 199 sarcoidosis patients, only 8 of them (4%) were infected with SARS-CoV2. The most frequently reported symptoms were asthenia (62%), fever (62%), and dysgeusia (62%), while about 50% of patients complained of myalgia, chest pain, diarrhea, and headaches. Moreover, three patients were admitted to the hospital and two to the intensive care unit, while one patient died (12.5%) [70]. A US-based study enrolling 954 patients with pulmonary sarcoidosis from a register including 27,8271 patients with COVID-19 reported an excess risk of morbidity and mortality in the sarcoidosis cohort; however, the propensity-matched analysis showed that this derived from a higher burden of comorbid diseases and other risk factors for severe COVID-19 prevalent in the pulmonary sarcoidosis group, compared to controls [71]. No specific data are available on the impact of different sarcoidosis treatments that might, in turn, influence the prevalence of comorbidities. In another study by Baughman and co-workers, 5200 patients worldwide completed a self-reporting COVID-19 questionnaire. The mean age of the study population was 54 years; 116 patients reported being infected, while 18 of them (16%) required hospitalization. The risk of COVID-19 infection was higher among patients with pulmonary or neurological disease and on rituximab treatment, while patients with cardiac disease were at a higher risk of hospitalization. Notably, no specific treatment was found to be associated with a higher risk of hospitalization. [72]. The main findings of the above-mentioned studies are recapitulated in Table 1. Interestingly, several authors reported the occurrence of pulmonary sarcoidosis following COVID-19 pneumonia, suggesting a possible link between the two diseases [10,73]. However, the mechanisms through which granulomatous inflammation develops following SARS-CoV2 exposure remain unknown.

## 4. Diagnosis of Sarcoidosis and COVID-19, the Role of Imaging

### 4.1. High Resolution Computed Tomography Findings (HRCT) of Pulmonary Sarcoidosis

HRCT is more sensitive than chest x-ray for thoracic manifestations of sarcoidosis and is very useful for a differential diagnosis between active and reversible inflammation (micronodules, ground-glass, and alveolar opacities) from irreversible fibrosis (volume loss, honeycombing, architectural distortion, and traction bronchiectasis) [74,75].

A perilymphatic distribution of small (2–4 mm diameter) and well-defined micronodules (corresponded to granulomas) is the most common parenchymal disease pattern. The micronodules, localized along the peribronchovascular and subpleural interstitial space and interlobular septa (Figure 1, Figure 2a, and Figure 3a), are characterized by a bilateral and symmetric distribution, prevalent in the upper lobes [76]. Another typical feature of sarcoidosis is the hilar and mediastinal lymph node enlargement, usually symmetric and bilateral (Figure 2a) [77,78].

A wide variety of atypical manifestations are solitary nodules (1–4 cm diameter), conglomerate masses (“galaxy sign”) (Figure 2b and Figure 3b), unilateral or isolated lymphadenopathy, patchy airspace consolidation, patchy ground-glass opacities (represent extensive interstitial sarcoid granulomas below the resolution of HRCT rather than alveolitis), miliary opacities [79,80], linear reticular opacities, airway involvement with air trapping, tracheobronchial abnormalities and atelectasis, and finally, pleural disease with effusions, thickening, calcification, or pleural plaque-like opacities [78].

The advanced stages of the disease (stage IV) are characterized by fibrocystic changes with the presence of honeycomb-like cysts, traction bronchiectasis, upward and outward retraction of hila, compensatory hyperinflation of the lower lobes, cavitation of parenchymal lesions, and mycetoma formation [81].

There are three main patterns known to correspond to pulmonary function test results.

Obstructive physiology: characterized by bronchiectasis featuring air trapping or central bronchial distortion.Restrictive physiology with low DLCO: characterized by subpleural honeycombing.Mild effect on respiratory function: characterized by diffuse linear fibrotic pattern, with typical distribution more from hila to all directions [82].

### 4.2. HRCT Findings of Lung Involvement from COVID-19

Unenhanced CT is considered the main investigator for diagnosis, detection of complications, and prognostication of COVID-19 because it allows more sensitive results than the chest X-ray [83]. More than half of patients imaged quickly after symptom onset have a normal CT scan.

Chest CT patterns with a high incidence that are hallmark CT manifestations of COVID-19 are represented by peripheral ground-glass opacities (early stage; 0–5 days after symptom onset) (Figure 3c) with or without lung consolidation (peak stage; 9–13 days after symptom onset) (Figure 3c,d) sometimes with a rounded morphology and lower lobe and posterior predilection [84].

Chest CT patterns with intermediate incidence are represented by crazy-paving pattern (Figure 3c,d), linear opacities, reverse halo sign, and signs of fibrosis (late stage; ≥14 days after symptom onset) with the presence of traction bronchiectasis, parenchymal bands, and architectural distortion [84].

### 4.3. Differences and Similarities between HRCT Findings of Pulmonary Sarcoidosis and COVID-19

While the perilymphatic distribution of small micronodules and hilar and mediastinal lymph node enlargement is the hallmark CT manifestations of pulmonary sarcoidosis, they are characteristically absent in COVD-19 disease (Figure 3). Furthermore, ground-glass opacities, representing the CT hallmark of COVID-19 disease, are rare and nonspecific findings of pulmonary sarcoidosis [85].

In the first wave of the pandemic, it was observed that while pulmonary sarcoidosis mainly affected the upper lobes, COVID-19 pneumonia was mainly distributed in the lower lobes. The more recent SARS-COV-2 Omicron variant was associated with fewer and less severe CT alterations, reduced clinical severity, and hospitalization risk, as compared to the Delta variant [86].

Both pathologies can cause cardiac involvement. While cardiac sarcoidosis is characterized by the presence of myocyte necrosis (acute phase) and interstitial fibrosis (chronic phase) with both non-ischemic (intramyocardial or subepicardial) and ischemic (subendocardial or transmural) patterns in more than one coronary territory, peri-myocarditis following Coronavirus Disease 2019 (COVID-19) revealed a typical mid-subepicardial non-ischemic pattern mainly at the left ventricle lateral wall [87,88,89,90,91].

## 5. Treatment Approaches, Algorithms for Suppression of Sarcoidosis during the COVID-19 Pandemic

Large epidemiological studies conducted on data from the first COVID-19 pandemic wave in the United States suggest that patients with sarcoidosis have an increased risk of developing severe forms of COVID-19 and mortality [92]. This risk was particularly pronounced in those patients in which sarcoidosis was associated with moderate or severe impairment of lung function [93]. A large multi-center investigation conducted on patients with rheumatic diseases, including sarcoidosis, demonstrated that treatment with ≥10 mg of prednisolone equivalent per day was the independent risk factor for COVID-19-related hospitalization and mortality, not disease severity per se [94].

Based on these findings, experts on the treatment of rheumatic diseases have issued prudent consensus recommendations on the management of patients undergoing treatment for sarcoidosis during the COVID-19 pandemic [95,96].

These recommendations are summarized in Table 2. The basic concept is to limit immunosuppressive treatments and their doses to the strictly necessary ones to maintain disease stability or control the active disease. Namely, in patients following glucocorticoid treatments only, doses should be reduced to the lowest possible or lowest effective to control active sarcoidosis. For all other drugs, the risks and benefits of treatment continuation should be weighed before taking any therapeutical decision, and reduction of doses or prolongation of dose intervals should be considered for biological drugs.

The prudent attitude of these recommendations is justified by the substantial novelty of SARS-CoV-2 infection and a lack of deep knowledge of its pathophysiological mechanisms from one side, and by the assumption that prolonged treatment with glucocorticoid or biological drugs can increase the risk of severe infections, as observed in many other rheumatic diseases, from the other [97]. However, these recommendations were issued on expert consensus in the absence of any study specifically investigating the optimal anti-sarcoidosis treatment regimen for minimizing the risk of severe COVID-19. The extrapolation of evidence collected in the context of another rheumatic disease to sarcoidosis may not necessarily lead to correct conclusions [98].

Furthermore, the relationship between chronic immunosuppressant treatments and COVID-19-related outcomes is still far from understood from an epidemiological perspective [99]. Intravenous administration of glucocorticoids, namely dexamethasone, is currently one of the pharmacological mainstays for the treatment of COVID-19 pneumonia requiring hospital admission, for its capacity to reduce the uncontrolled activation of inflammatory response in the advanced phase of severe SARS-CoV-2 infection [100]. The anti-interleukin-6 biological drug tocilizumab has demonstrated some efficacy in the treatment of severe forms of COVID-19 [17]. Inhaled glucocorticoids also seem to have an important role in preventing the progression of COVID-19 to more severe forms [101]. In this context, chronic treatments against sarcoidosis may not necessarily be harmful to the clinical course of COVID-19, and future studies should specifically investigate this issue.

The current state-of-the-art literature suggests that in patients with sarcoidosis, the prognosis of COVID-19 is mainly driven by the presence of comorbidities, which are generally more prevalent in this particular population [71].

Older age, frailty, multimorbidity, and timing of contagion are, in fact, the main factors associated with adverse COVID-19 outcomes in the general population [102,103], and these associations probably remain valid also for patients with sarcoidosis [104].

In this scenario, the management of therapies for sarcoidosis should necessarily be tailored to each individual risk profile, with a reasonable compromise between the risk of severe forms of COVID-19 and the onset of acute flares of sarcoidosis.

As for other immune-mediated diseases, concerns and discussions have been raised regarding the timing and opportunity of anti-SARS-COV-2 vaccinations in sarcoidosis patients in relation to disease activity and ongoing treatment [105]. No specific studies have been published on sarcoidosis cohorts; one is currently recruiting (NCT05089565). Available data and recommendations are mainly derived from other systemic immune-mediated inflammatory diseases [106,107], suggesting vaccination administration prior to planned immunosuppression, if clinically possible, and adjustment of immunosuppressive therapy to optimize vaccination response during periods of well-controlled disease. Response to vaccination might be impaired by concomitant treatment [107]. However, also taking into account previous studies on different vaccines in sarcoidosis patients [108], vaccination against SARS-CoV-2 is strongly recommended in patients with sarcoidosis and could contribute to reducing the burden of COVID-19 severity once SARS-CoV-2 infection is on.

## 6. Conclusions

The complex interplay between sarcoidosis and COVID-19 remains a field of great discussion. The diffusion of an effective vaccination campaign has led to a significant improvement in clinical outcomes in patients affected by SARS-CoV-2 infection. However, the immune evasion from SARS-CoV-2 variants and waning immunity over time are two fearsome features that should be taken into consideration before considering easing virus containment measures. This aspect is particularly true in those patients at high risk of severe diseases, such as comorbid sarcoidosis patients. Due to similar immunopathogenetic clinical features, key points of future management will be the combined treatment of both conditions. It would be useful indeed to evaluate whether a combined therapy would add additional benefits to long-term outcomes, overall survival, and risk of hospitalization in patients affected both by sarcoidosis and COVID-19.

## Figures and Tables

**Figure 1 biomedicines-10-02525-f001:**
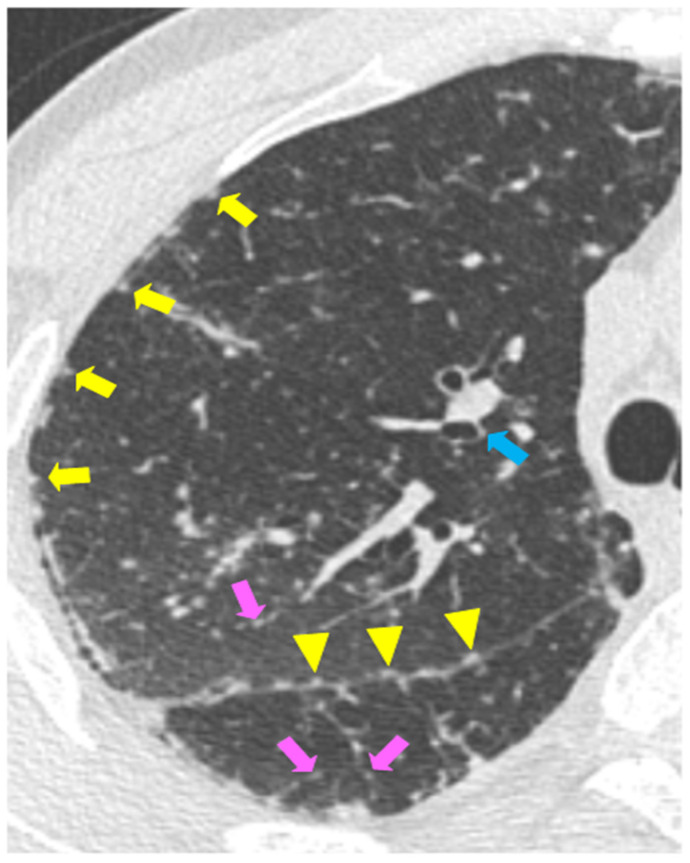
Distribution of micronodules in sarcoidosis. The axial HRCT scan in a patient with pulmonary sarcoidosis shows the typical perilymphatic distribution of micronodules along the subpleural interstitial space (yellow arrows), along the fissure (yellow arrowheads), and interlobular septa (pink arrows). The blue arrow shows the peribronchovascular distribution.

**Figure 2 biomedicines-10-02525-f002:**
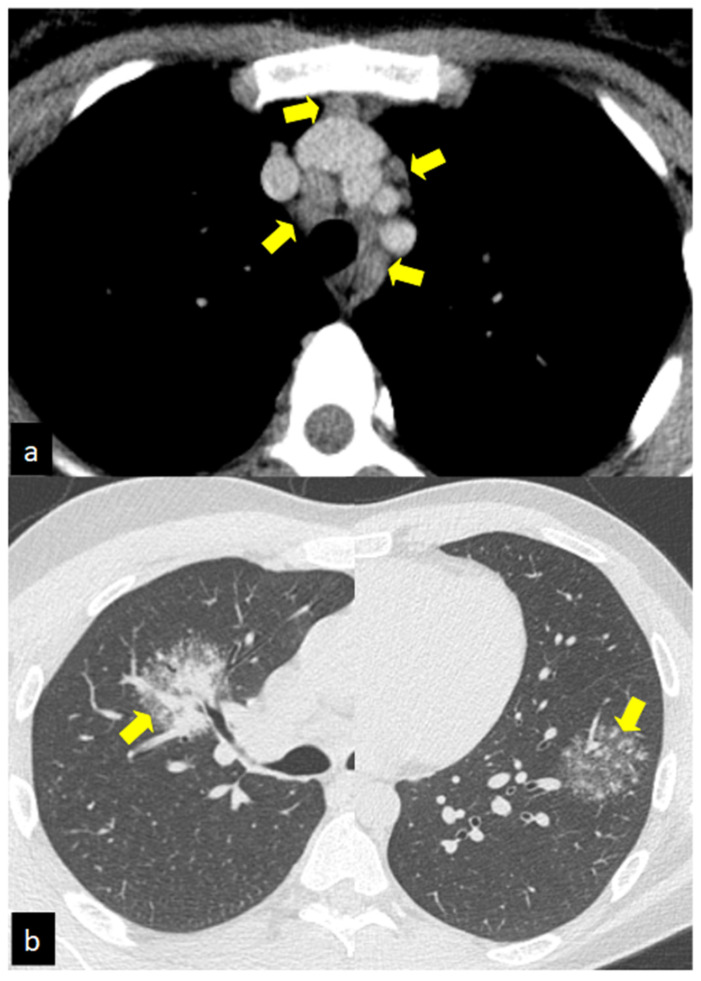
Typical and atypical HRCT manifestations of sarcoidosis. Hilar and mediastinal lymph node enlargement is usually symmetric and bilateral in sarcoidosis (yellow arrows, **a**). Atypical manifestations of sarcoidosis: solitary nodules (1–4 cm diameter), conglomerate masses (“galaxy sign”) (yellow arrows, **b**).

**Figure 3 biomedicines-10-02525-f003:**
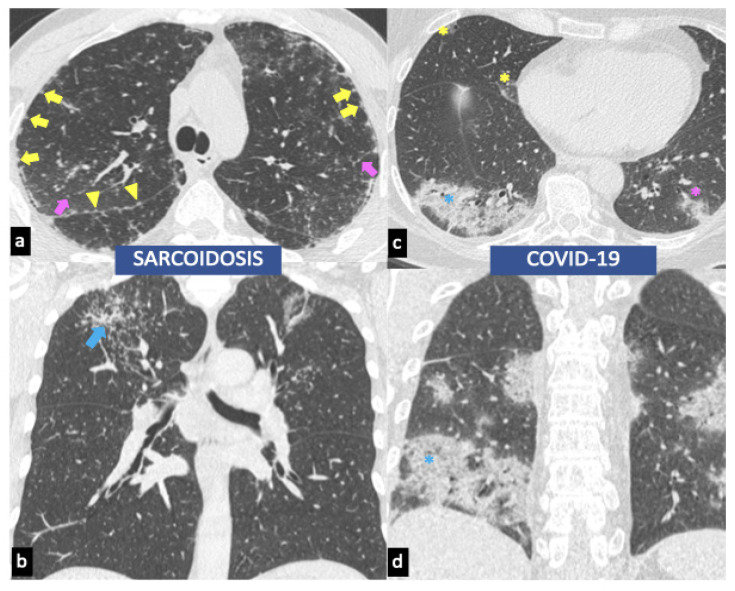
HRCT findings of sarcoidosis vs. COVID-19. Sarcoid-related micronodules corresponding to granulomas have a perilymphatic distribution and are located along scissural (yellow arrowheads), subpleural interstitial space (yellow arrows), and interlobular septa (pink arrows) in a symmetric and upper lobe distribution (**a**). Galaxy sign is an atypical finding of sarcoidosis (blue arrow, **b**). Peripheral ground-glass opacities (early stage; 0–5 days after symptom onset) are instead frequent HRCT manifestations of COVID-19 in the pre-vaccination era (yellow asterisks, **c**), with or without lung consolidation (peak stage; 9–13 days after symptom onset, pink asterisks, **c**). A crazy-paving pattern has an intermediate incidence (blue asterisks, **c**,**d**).

**Table 1 biomedicines-10-02525-t001:** Studies reporting the impact of SARS-CoV-2 infection in sarcoidosis patients.

Country	Spain [67]	US [68]	US [69]	France [70]	US [71]	All over the world [72]
Initial cohort	882 (calculated) sarcoidosis patients	236 sarcoidosis patients	899 sarcoidosis patients	199 sarcoidosis patients	278,271 COVID patients, of which 954 with pulmonary sarcoidosis	5200 sarcoidosis patients (self-reported questionnaire)
Time	2020	12 March to 30 April 2020	2nd semester 2020	Feb–May 2020	Jan–Oct 2020	
Prevalence of infection	5.1% (45 patients)	2.1% (5 patients)	8.9% (77 patients)	4.02% (8 patients)	N/A	2.23% (116 patients)
Symptomatic	36/45 (80%)	5/5 (100%)	Not specified	8 (100%)	N/A	
Hospitalized	14/45 (31%)	2/5 (40%)	19 (25%)	3/8 (37.5%)	181/954 (18.87%)	18/116 (15.8%)
ICU	2/45 (4.4%)	2/5 (40%)	6/19 (31%)	2/8 (25%)	66/954 (6.9%)	Unknown
Died	4/45 (9%)	1/5 (20%)	1/77 (1%)	1/8 (12.5%)	41/954 (4.29%)	N/A
Relation treatment-severity?	Not evaluated	Not evaluated	Not evaluated	Not evaluated	Not evaluated	None
N/A = not applicable						

**Table 2 biomedicines-10-02525-t002:** Overview of expert recommendations on the optimal management of sarcoidosis treatment during the COVID-19 pandemic.

Clinical Scenario	Management during the COVID-19 Pandemic
Stable disease not requiring drug treatment	No treatment
Stable disease under glucocorticoids only	Reduction to the lowest effective dose
Stable disease under DMASD	Reduction of doses, prolongation of intervals, possible drug holiday
Stable disease under biologicals	Reduction of doses, prolongation of intervals
Non-life-threatening disease flare	Increase glucocorticoid doses to the lowest effective dose
Active disease under glucocorticoids only	Reduction to the lowest effective dose
Active disease under DMASD	Continuation of DMASD only if benefits outweigh risks of severe COVID-19
Active disease under biologicals	Continuation of treatment (different routes are possible during lockdowns)
Continued active disease	Consideration of alternative treatments

DMASD = Disease-Modifying Anti-Sarcoid Drugs

## Data Availability

Not applicable.

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
