# Peer review of "Sarcoidosis and COVID-19: At the Cross-Road between Immunopathology and Clinical Manifestation"

_biomedicines, 2022, doi:10.3390/biomedicines10102525_

Round 1

Reviewer 1 Report

This review summarizes the relationship between sarcoidosis and COVID-19 and provides an overview on the immunopathology, clinical and diagnostic features. Overall, the manuscript is well written and addresses various aspects of the complex interplay between sarcoidosis and COVID-19.

I do not have any major concerns, but I think that the quality of the manuscript would benefit if the authors write a paragraph on the interplay between complement and sarcoidosis and complement and COVID-19 in their section ‘Innate and adaptive immunity in COVID-19 and sarcoidosis’ and also discuss these findings in the Discussion.

Also, the paragraph on vaccination is very short, maybe the authors could elaborate more on this topic and also include if anything is known in the context of infection with SARS-CoV-2 variants of concern in these patients.

Author Response

Rev1 This review summarizes the relationship between sarcoidosis and COVID-19 and provides an overview on the immunopathology, clinical and diagnostic features. Overall, the manuscript is well written and addresses various aspects of the complex interplay between sarcoidosis and COVID-19.

Rev1: I do not have any major concerns, but I think that the quality of the manuscript would benefit if the authors write a paragraph on the interplay between complement and sarcoidosis and complement and COVID-19 in their section ‘Innate and adaptive immunity in COVID-19 and sarcoidosis’ and also discuss these findings in the Discussion.

We thank Rev1 for her his suggestions. Accordingly, the manuscript has been updated.

Humoral innate immunity:

The role of humoral innate immunity in SARS-CoV-2 response has been deeply investigated, showing that specific humoral fluid-phase pattern-recognition molecules (PRMs) with ancestral antibody-like properties, including collectins, ficolins, pentraxins and C1q, may play a role both in resistance to and in the pathogenesis of COVID-19 [14]. Circulating and lung myelomonocytic cells and endothelial cells have been reported as a major source of the pentraxin long pentraxin 3 (PTX3) in COVID-19 [104]. PTX3 is a PRM playing essential functions in resistance to pathogens, tissue remodeling and resolution of inflammation. Notably, in COVID-19, its plasma concentration has been suggested as an independent strong prognostic indicator of short-term mortality. Interstingly, different reports have hypothesized mechanistic links between the functional activity of humoral innate immunity and sarcoidosis pathogenesis. A significant enrichment in complement-activating factors in bronchoalveolar lavage (BAL) of sarcoidosis patients and increased alveolar concentrations of C5a were detected compared to other interstitial lung diseases. An increased expression of complement receptors has also been reported in monocytes from sarcoidosis patients [105]. Recently, it has been demonstrated that PTX3, in particular, may restrain the pathogenic activation of complement and its downstream impact on metabolic reprogramming of macrophages in sarcoidosis, thus acting as a physiological brake in granuloma formation [106].

Rev1: Also, the paragraph on vaccination is very short, maybe the authors could elaborate more on this topic and also include if anything is known in the context of infection with SARS-CoV-2 variants of concern in these patients.

Vaccination:

As for other immune-mediated diseases, concerns and discussions have been raised regarding timing and opportunity of anti-SARS-COV-2 vaccinations in sarcoidosis patients, in relation to diseases activity and ongoing treatment [100]. No specific studies have been published on sarcoidosis cohorts, one is currently recruiting (NCT05089565). Available data and recommendations mainly derive from other systemic immune-mediated inflammatory diseases [101, 102] [Reference ACR and Simon et al.] suggesting vaccination administration prior to planned immunosuppression, if clinically possible, and adjustment of immunosuppressive therapy to optimize vaccination response during periods of well-controlled disease. Response to vaccination might be impaired by concomitant treatment [102]. However, taking also into account previous studies on different vaccines in sarcoidosis patients [103, Ref Tavana et al], vaccination against SARS-CoV-2 is strongly recommended in patients with sarcoidosis, and could contribute to reducing the burden of COVID-19 severity once SARS-CoV-2 infection is on[100].

Reviewer 2 Report

Overall the review discusses several important key points in the comparison between the two diseases and their possible interactions. However, several major points need more attention. The claim that “sarcoidosis are at high risk of severe COVID” in the abstract is not substantiated in the text. Indeed, in lines 208-209 the authors state that “sarcoidosis patients seem to have the same risk of infection of the general population but with a higher risk of severe disease”, while in lines 396-397 they claim that “they may have an increased risk of contracting SARS-CoV-2 infection” On the same problem, the paragraph in lines 217~233 is very confuse. It would be very informative to have a figure that summarizes the available data on infection rates and severity (hospitalization, death…), possibly in relation to immunosuppressive therapy. There are sparse references to drugs used in COVID therapy throughout the text. The sentence in lines 127-128 needs to be more precise: what is the result of these therapies?stating that they have been used is not much of a demonstration. Moreover, IL-6 therapy is also mentioned, with a different reference, in line 391. As a final comment, the authors should carefully check the references. For instance, Ref 21 and 26 seem to be wrong.

Author Response

Rev2

Rev2: Overall the review discusses several important key points in the comparison between the two diseases and their possible interactions. However, several major points need more attention. The claim that “sarcoidosis are at high risk of severe COVID” in the abstract is not substantiated in the text. Indeed, in lines 208-209 the authors state that “sarcoidosis patients seem to have the same risk of infection of the general population but with a higher risk of severe disease”, while in lines 396-397 they claim that “they may have an increased risk of contracting SARS-CoV-2 infection” On the same problem, the paragraph in lines 217~233 is very confuse. It would be very informative to have a figure that summarizes the available data on infection rates and severity (hospitalization, death…), possibly in relation to immunosuppressive therapy. There are sparse references to drugs used in COVID therapy throughout the text. The sentence in lines 127-128 needs to be more precise: what is the result of these therapies?stating that they have been used is not much of a demonstration. Moreover, IL-6 therapy is also mentioned, with a different reference, in line 391. As a final comment, the authors should carefully check the references. For instance, Ref 21 and 26 seem to be wrong.

Whe thank Rev2 for her/his suggestions. Accordingly, we modified the text.

Lines 208-209 and 213-233: section has been edited according to Rev2 suggestions and a new table (Table 1) has been added (see below).

Clinical features of sarcoid patients infected by SARS-CoV2

Little is known about the long-term consequences of SARS-CoV2 infection in patients with preexisting chronic lung diseases. Only a few studies have analyzed the impact of SARS-CoV2 infection in patients with sarcoidosis and data are limited to small case series or isolated case reports. With these limitations, sarcoidosis patients seem to have the same risk of infection as the general population people but with a higher risk of severe disease. In a study from Spain, the prevalence of SARS-CoV2 infection in patients with sarcoidosis was 5.1%, with 80% of cases being symptomatic. The most frequently reported symptoms are cough (67%), fever (61%), fatigue (42%), and myalgias (28%); conversely, only a minority of patients had dyspnea (22%)[62]. Moreover, 31% of patients were hospitalized and 9% died following COVID-19 infection. During the first pandemic wave, a similar prevalence was reported in a small case series of African American patients in the US (2.1%)[63].

In a large cohort of sarcoidosis patients (n=886), Baughman and colleagues reported a hospitalization rate of 25% and only one death from COVID-19[64]. Furthermore, no individual immunosuppressive therapy was associated with an increased risk of SARS-CoV2 infection. Similarly, in a French cohort of 199 sarcoidosis patients, only 8 of them (4%) were infected with SARS-CoV2. The most frequently reported symptoms were asthenia (62%), fever (62%), and dysgeusia (62%) while about 50% of patients complained of myalgia, chest pain, diarrhea, and headaches. Moreover, three patients were admitted to hospital and two to intensive care, while one patient died[65]. In another study by Baughman and co-workers, 5.200 patients worldwide completed a self-reporting COVID-19 questionnaire. The mean age of the study population was 54 years; 116 patients reported being infected, while 18 of them (16%) required hospitalization. The risk of COVID-19 infection was higher among patients with pulmonary or neurological disease and on rituximab treatment, while patients with cardiac disease were at a higher risk of hospitalization[66]. The main findings of the above-mentioned studies are recapitulated in Table 1. Interestingly, several authors reported the occurrence of pulmonary sarcoidosis following COVID-19 pneumonia, suggesting a possible link between the two diseases[10, 67]. However, the mechanisms through which granulomatous inflammation develops following SARS-CoV2 exposure remains unknown.

Table 1: studies reporting impact of SARS-CoV-2 infection in sarcoidosis patients

SPAIN (62)

US (63)

Baughman, U.S. (64)

France (65)

U.S. (107)

All over the world (66)

Initial cohort

882 (calculated) sarcoidosis patients

236 sarcoidosis patients

899 sarcoidosis patients

199 sarcoidosis patients

278271 COVID patients of which 954 with pulmonary sarcoidosis

5200 sarcoidosis patients (self reported questionnaire)

Time

2020

March 12 to April 30, 2020

2nd semester 2020

Feb-May 2020

Jan-Oct 2020

Prevalence of infection

5.1% (45 patients)

2,1% (5 patients)

8.9% (77 patients)

4.02% (8 patients)

N/A

2.23% (116 patients)

Symptomatic

36/45 (80%)

5/5 (100%)

Not specified

8 (100%)

N/A

Hospitalized

14/45 (31%)

2/5 (40%)

19 (25%)

3/8 (37.5%)

181 (18.87%)

18/116 (15.8%)

ICU

2/45 (4.4%)

2/5 (40%)

6/19

2/8 (25%)

66 (6.9%)

Unknown

Died

4/45 (9%)

1/5 (20%)

1/77

1/8 (12.5%)

41 (4.29%)

N/A

Relation treatment-severity?

Not evaluated

Not evaluated

Not evaluated

Not evaluated

Not evaluated

None

N/A= not applicable

Lines 396-397: sentence has been modified, in line with previous changes.

Lines 127-128:. the sentence (now lines 161-162) has been modified “anti-IL-1b and anti-IL-6 monoclonal antibodies, as well as JAK-inhibitors have been approved for COVID-19 treatment in hospitalized patients, in order to underline that these drugs have been authorized for use in COVID-19 pneumonia and reference 94 has been added.

Finally, references have been checked.
